# What Is the Relevance in the Passing Action between the Passer and the Receiver in Soccer? Study of Elite Soccer in La Liga

**DOI:** 10.3390/ijerph17249396

**Published:** 2020-12-15

**Authors:** Antonio Cordón-Carmona, Abraham García-Aliaga, Moisés Marquina, Jorge Lorenzo Calvo, Daniel Mon-López, Ignacio Refoyo Roman

**Affiliations:** Facultad de Ciencias de la Actividad Física y del Deporte (INEF-Departamento de Deportes), Universidad Politécnica de Madrid, C/Martín Fierro, 7, 28040 Madrid, Spain; antoupm@gmail.com (A.C.-C.); abraham.garciaa@upm.es (A.G.-A.); moisesmn95@gmail.com (M.M.); ignacio.refoyo@upm.es (I.R.R.)

**Keywords:** performance indicators, performance analysis, tactical behavior, soccer

## Abstract

Soccer is a high-complexity sport in which 22 players interact simultaneously in a common space. The ball-holder interacts with their teammates by passing actions, establishing a unique communication among them in the development of the game in its offensive phase. The main aim of the present study was to analyze the pass action according to the trajectory of the ball receiver and the space for receiving the ball in terms of success at the end of play. Twenty La Liga 2018/2019 matches of two elite teams were analyzed. A system of notational analysis was used to create 11 categories based on context, timing and pass analysis. The data were analyzed using chi-squared analysis. The results showed that the main performance indicators were the efficiency of the pass, the zone of the field, the trajectory of the receiver and the reception space of the ball, which presented a moderate association with the end of play (*p* < 0.001). We concluded that receiving the ball on approach and in separation increased the probability of success by 5% and 7%, respectively, and a diagonal run increased the probability by 7%. Moreover, the combined analysis of these variables would improve the team performance.

## 1. Introduction

The search for theoretical tactical models for soccer is gaining importance in the current scientific literature. The creation of a theoretical tactical model would provide the head coach with the appropriate information for optimal match preparation [1,2,3]. More specifically, Low et al. [2] defined tactics as the actions taken by players in adaptation to dynamically changing match situations. Similarly, Kempe et al. [4] observed that team tactics are governed by a complex process resulting from a combined network of interdependent factors. However, soccer is a highly complex sport [5]. This complexity is derived from the interaction among teammates and between opponents, the rules and the distribution in a common space to be navigated with the feet [6,7,8].

In this context, high-level soccer coaches are always looking to improve tactical performance and rely on the analysis of multiple variables, such as the playing structure of their team and the opponent, playing at home or away, the weather, the state of the game and the competition [3,9,10,11,12,13,14,15,16]. Among these factors, the observation technique has been used in the analysis of team sports to determine players’ behaviors and performance (both individually and collectively) in the overall team performance. This analysis is carried out through a notational system by a team member or coaching staff members [17]. In addition, this notational analysis looks at parameters that do not provide a concrete theoretical explanation of team behavior in the form of certain parameters that influence players and team performance [3]. 

In order to better understand notational analysis, the development of computer and video-assisted match analysis systems (such as Sportscode, Focus X2, ProZone and AMISCO) and technological innovations, such as global positioning systems (GPS) [18], vision systems and radio-based tracking systems [19], have helped to make data more accessible [13,20]. In this regard, Randers et al. [21] studied four different analysis systems, including two GPS systems, a video-based time–motion analysis system and a semi-automatic multicamera system. The results of this study showed similar decreases in performance in the four systems but there were significant differences among them related to absolute distances covered. 

However, tactical analysis in soccer has been based on the use of statistics and averages of decontextualized parameters, such as passing [19,21], ball possession [10,22,23], playing style [24,25,26,27] and ball recovery location [28,29]. Vogelbein et al. [30] evaluated the time required by German Bundesliga soccer teams in ball retrieval, showing that the time from a team’s loss of the ball to its recovery was a performance factor. This form of analysis rules out contextual information where other studies propose analyzing a set of variables to avoid that problem [2,31]. 

Other authors studied team tactics using pass networks; for example, they looked at the players who were most important within the team and the connections that were made among the players through passing [32,33,34]. Garrido et al. [35] used pass networks to calculate consistency and identifiability rankings over an entire season, for which Real Madrid CF was detected as having the highest consistency in their pass networks and FC Barcelona obtained the highest identifiability. Voronoi’s diagrams have also been used to analyze space dominance and overtaking of players in passing, resulting in a higher average for match wins when there were fewer of the opposing team’s players between the ball and the opponent’s goal and thus a larger area of the pitch [19]. On the other hand, other studies have analyzed the team’s centroid [2,7,36], the team’s dispersion [2,8,37] and team synchrony [2,32,38,39], showing how team members must carry out actions together to become a single body. 

Finally, the technical action of the pass has been analyzed from a qualitative perspective, assessing success and mistakes [19,34,40], and from a quantitative point of view, counting the number of passes [36,41]. However, this type of analysis does not take into account the player who receives the ball from their partner, this variable being an important performance aspect.

Based on the previous scientific literature, there seems to be a lack of research with regard to the analysis of the player who receives the ball, at least to the best of the author’s knowledge. In the study of only one of the members of the pass action, a lot of information is discarded from the communication between the players. The answers to the questions “where does he receive?” and “how does he receive?” provide us with relevant information about the pass receiver. Where and how the ball is received by the receiver of the pass can be explained by studying the trajectory and space in which the player receiving the pass takes possession. The play will perform better at its completion depending on how and where the receiver of the pass receives it. Therefore, the objectives of this study are: (1) to analyze the effectiveness of the pass concerning the passer and receiver of the ball as a function of the completion of the play; (2) to study the trajectory of the receiver of the ball and the space where they receive the ball as a function of the success in completing the play, (3) to examine the diagonality of the pass and the area of the field where it is made.

## 2. Materials and Methods 

### 2.1. Sample

To carry out the study, the pass variable was analyzed in 20 soccer matches of the 2018/2019 season of La Liga (Spanish first division soccer league) using video recordings. The selected matches were from matchday 29 to matchday 38 of the Real Madrid CF (RM) and FC Barcelona (FCB) teams. These matches were chosen given that the fight for the league title and qualification for European competitions presupposed a higher level of the teams. A total of 1399 plays were analyzed—788 were by RM and 813 by FCB—and 10,128 passes, of which 4463 were by RM and 5665 by FCB. A total of 202 plays and passes were eliminated due to the impossibility of observing the whole play due to the broadcast of the match. The passes were analyzed when the possession of the ball involved two or more passes received by the teammate. Plays that had a pass or two passes but one of them was not received by the teammate were not analyzed. Corner kick set pieces were not analyzed when a teammate took a short kick or cross. Analysis was concluded after the opponent touched the ball or when a new possession was initiated. Publicly available data were collected that did not require any formal approval by an institution. 

### 2.2. Procedures

The data were examined using a modified notational system from Sarmento et al. [42]. The field was divided into 20 zones, modifying the reference model of Fernández-Navarro et al. [43] (Figure 1). Zones 1 to 5 were called defensive zone, 6 to 10 pre-defensive zones, 11 to 15 pre-offensive zones and 16 to 20 offensive zones. The score pentagon was included where there was a higher probability of scoring [44].

An experienced senior match analyst recorded all 20 soccer matches. Intra-observer reliability was achieved by repeating the notational analysis of two matches (one from each team) at random after 15 days, without observing any matches to avoid effects on the memory of the analysis.

### 2.3. Categories and Dimensions

The pass variable has been analyzed according to different categories and dimensions in previous research [10,42,44,45]; here it was carried out following the observational method proposed by Anguera et al. [46] (Table 1).

The concept of the diagonality of the pass was established as the direction in which the pass was directed depending on the area of the pitch and the place where the team was attacking or defending. With regard to the receiver’s trajectory, this was defined as the direction of the run where the player who was going to receive the pass was heading for in the interaction space. On the other hand, the pass-receiving space was established as the reference place between two players in the defensive phase, the direct opponent of the strong side being next to the closest defender to where the ball was directed, tracing an interval between the two when the pass receiver took control of the ball. All the passes were analyzed according to the zone in which they occurred (Figure 1). 

The concept of a shot to the center of the area was defined as any ball sent to the opposing team’s area from a side of the field, which may be a pass of an elevated form or not [12]. The effectiveness of the pass was differentiated as bad, good or very good, given that differentiating it into three categories was the most appropriate way to avoid differences [40]. 

### 2.4. Data Analysis

Descriptive data for ordinal variables are presented as mean (*M*) and standard deviation (*SD*) and those for nominal variables as percentages. The Kolmogorov-Smirnov test was used to determine whether a data set was well-modeled by a normal distribution and to compute how likely it was for a random variable underlying the data set to be normally distributed. The Kappa index [47] was used to analyze intra- and inter-observer reliability matching of three observers. Chi-squared tests for nonparametric data were carried out to analyze whether completion efficiency was independent of the variables analyzed. Inter-observer reliability was achieved by analyzing the observations of two secondary observers of one match at random and comparing them with those of the main analyst. As an index of the effect size, Cramer’s V (Vc) was calculated and its interpretation was based on the following criteria: Vc < 0.20 weak association, 0.20 ≤ Vc < 0.40 moderate, 0.40 ≤ Vc < 0.60 relatively strong and Vc ≥ 0.60 strong association [48]. Data analysis was performed with IBM SPSS version 25.0 for Windows (IBM Corporation, Armonk, NY, USA). The significance level was set at 0.05.

## 3. Results

The results obtained by the observation instrument refer to data quality control, focusing on intra- and inter-observer matching, and are shown in Table 2.

### About the Effectiveness of Completion 

The pass efficiency presented a moderate association in its totality (χ^2^ (4) = 851.785; *p* < 0.001; Vc = 0.205) and in the plays of 2 to 8 passes. Moderate association was also shown in plays of 10 and 15 passes. It should be noted that the play with three passes had the greatest association (χ^2^ (4) = 209.027; *p* < 0.001; Vc = 0.382) (Table 3, Table 4 and Table 5 and Figure 2, Figure 3 and Figure 4).

With regard to the trajectory of the ball receiver, it can be seen to have a moderate association with two passes (χ^2^ (6) = 34.554; *p* < 0.001; Vc = 0.280) and three passes (χ^2^ (6) = 58.104; *p* < 0.001; Vc = 0.201). In plays with two passes, the run made diagonally obtained 44.5% efficiency and 31.8% for finishing the play successfully when it was received statically; when the play had three passes, it obtained 43.3% success when the run was diagonal and 30% efficiency when it was received statically. In the rest of the plays with a greater number of passes, there was little or no association, and none in their totality (χ^2^ (6) = 123.406; *p* < 0.001; Vc = 0.078). When the player received the ball in a static position, he had a 58.4% success rate. However, the effectiveness of the diagonal shift was 22.2%.

The space where the pass was received presented a moderate association when two passes (χ^2^ (4) = 42.634; *p* < 0.001; Vc = 0.311) and three passes (χ^2^ (4) = 86.653; *p* < 0.001; Vc = 0.246) were made; in the other plays with other numbers of passes there was a weak association. When the play had two passes, it was 39.1% successful in a receiving position and receiving a pass in separation was 41.8% effective. In the case of plays with three passes, 41.7% was achieved in the two places where the ball was received. In total, there was a weak association (χ^2^ (4) = 224.671; *p* < 0.001; Vc = 0.105). When the play was received positively, there was a 70.6% success rate. Conversely, receiving a ball in separation obtained a 16.4% success rate.

In the combination of the trajectory and the space where the receiver received, there was a moderate association among the two-pass plays (χ^2^ (18) = 51.641; *p* < 0.001; Vc = 0.343), three-pass plays (χ^2^ (22) = 105.869; *p* < 0.001; Vc = 0.272), four-pass plays (χ^2^ (22) = 74.667; *p* < 0.001; Vc = 0.221) and seven-pass plays (χ^2^ (22) = 62.613; *p* < 0.001; Vc = 0.214). When the play was of two passes, the reception of the passes diagonally in separation achieved 30% success and the reception in a static position 29.1% success. When the play had three passes, the efficiency of the reception of a pass diagonally was 30% and in a static position 27.8%. When the play was made with four passes, the reception success was 22.9% with a diagonal pass and 41% in a static position. In the case of plays with seven passes, there was a 13% success rate in the reception of a diagonal pass and 48.1% in a static position. However, in plays with other numbers of passes, a weak association was denoted, as in the total data (χ^2^ (22) = 248.305; *p* < 0.001; Vc = 0.111), leading to 55.5% of the play finishing successfully when a pass was received in a static position followed by diagonal reception in separation with 12.2% efficiency.

The diagonality of the pass had a weak association, both in the totality of the data (χ^2^ (8) = 25.515; *p* < 0.01; Vc = 0.035) and in the different plays. The highest percentage obtained for finishing the play effectively by making a diagonal pass forward was 45.4%, followed by the diagonal pass backward with 26.2%.

In the combination of the trajectory and the diagonality of the pass, there was a moderate association in the play of 2 to 5 passes, 7 passes, 10 passes and 12 to 15 passes. However, weak association was evident in the play of 6, 8, 9 and 11 passes. In its totality, this combination presented no association (χ^2^ (38) = 172.232; *p* < 0.001; Vc = 0.092). Receiving a diagonal pass from behind in a static position led to 20.4% of the play finishing effectively, 18.7% when receiving a diagonal pass from the front in a static position and 15.6% success when receiving a pass from the front diagonally. When the play had two passes it had a percentage of finishing the play effectively of 28.2% when a diagonal pass forward was received in a diagonal run, followed by an 11.8% for a vertical run with a diagonal pass forward and 10.9% for a diagonal run with a pass forward. When the play involved three passes received on a diagonal run with a forward diagonal pass, 32.2% efficiency was achieved, followed by the vertical run with a forward diagonal pass with 12.8%. From pass three onwards the diagonal run with the forward diagonal pass reduced its percentage of finishing the play effectively in favor of receiving in a static position with the forward diagonal pass and the backward diagonal pass.

In the combination of the reception space and the diagonality of the pass, there was a moderate association of two to seven passes when the play had 15 passes. There was little association for 10 to 14 passes; with no significant difference in its totality, it presented little association (χ^2^ (28) = 291.749; *p* < 0.001; Vc = 0.120), leading to 25.2% effectiveness in finishing the play when the positional ball was received from a diagonal pass ahead. In addition, receiving a positional backward diagonal pass achieved a 23.1% success rate and receiving a separate forward diagonal pass a 12.9% success rate. When the play involved two passes, the reception in separation of a forward diagonal pass had a higher percentage of 27.3%, followed by receiving a positional forward diagonal pass in separation (15.5%), then a forward pass (10.9%) and, by approximation, a forward diagonal pass (10%.) When the play involved three passes, the reception in separation of a diagonal pass forward represented a 32.2% success rate, followed by the positional reception of a diagonal pass forward with 15.6%. When the play involved four passes, a higher percentage was obtained when receiving in a forward position at 25.2%, followed in this case by receiving a forward pass in separation (22.3%) and then positional reception with a diagonal backward pass (16.5%).

The results show how the area of the field from which the pass was made presented a moderate association with the completion of the play in all its passes, with a significant difference in the plays in 2 to 8 passes, 10 and 14 passes (*p* < 0.001), 9 passes (*p* < 0.05), 11 passes (*p* = 0.002) and in 15 passes (*p* = 0.001). However, in 12 and 13 passes there were no significant differences (*p* > 0.05). The highest association was obtained with the plays involving two passes (χ^2^ (46) = 65.471; *p* = 0.031; Vc = 0.386), which had the highest percentage of completion with efficiency in zone 13 of the field (pre-offensive zone in its central part), with 15.5%. In the total data there was a weak association (χ^2^ (48) = 480.484; *p* < 0.001; Vc = 0.154) (Table 4 and Table 5).

The influence of the final result of the match on the effectiveness of the completion of the play had a relatively strong association when the play had 15 passes (χ^2^ (8) = 144.949; *p* < 0.001; Vc = 0.518). When winning by more than one goal, finishing the play without efficiency occurred in 44.4% of cases and with efficiency in 42.9%. Conversely, when a team lost by one goal, the play only ended up neutral (n = 15). On the other hand, the final result had a moderate association when the play had 2, 7 or 9 to 14 passes. However, in its totality the final result of the match showed no association with the completion of plays (χ^2^ (8) = 123.189; *p* < 0.001; Vc = 0.078). When the final result of the match was positive, a greater percentage of the plays were effectively completed.

The time of the match had no association with the completion of plays except when the play had 13 passes, which showed a moderate association (χ^2^ (2) = 38.255; *p* < 0.001; Vc = 0.275). Ending the play effectively in the first half of the match demonstrated 38.5% efficiency and in the second half 61.5%. These results were the opposite when the play ended without efficiency. In its entirety, the time of the match showed no association with the completion of plays (χ^2^ (2) = 12.800; *p* < 0.01; Vc = 0.036). The success rate was similar when the play ended without efficiency, as neutral, or with efficiency.

The match location had a relatively strong association when the play involved two passes (χ^2^ (2) = 38.103; *p* < 0.001; Vc = 0.416). When the team analyzed played at home, finishing the play without success had a 61.3% efficiency. However, 63.6% efficiency was obtained when playing as the away team. On the other hand, the match location showed a moderate association when the plays had 11, 12 and 13 passes because when playing at home the team finished the play effectively at a higher percentage than when playing as the away team. In the total results, the match location showed no association (χ^2^ (2) = 73.421; *p* < 0.001; Vc = 0.085). When the match was on the home pitch of the team, the play ended effectively 56.6% of the time, and on their rival’s pitch it ended ineffectively43.4% of the time; 49.4% of the matches were played at home and 50.6% away.

The classification of the opponent showed a moderate association when plays involved 2 passes, 7 passes and 11 to 15 passes. However, in its totality it showed no association (χ^2^ (6) = 129.589; *p* < 0.001; Vc = 0.080).

The match status had a relatively strong association when plays involved13 passes (χ^2^ (8) = 174.935; *p* < 0.001; Vc = 0.415). On the other hand, it showed a moderate association in plays of 2, 7, 9, 10, 11, 12, 14 and 15 passes. There was no association in its entirety (χ^2^ (8) = 101.578; *p* < 0.001; Vc = 0.071).

## 4. Discussion

The main aim of this study was to analyzes passes with regard to the trajectory made by the receiver of the ball and the space where the player received it in relation to the closest direct defender and his efficiency in finishing the play. Previous related literature has not explored these aspects; we only found work on the importance of gaining space when receiving the ball [19] and pass networks [32,33,34,35]. Thus, important information was provided by this study about where and how the ball receiver should move. 

The main data concerned the relationship between the trajectory of the receiver and the space where he received the ball and suggested that if the receiver took possession of the ball in separation, it improved the probability of success by 7% and by approximation by 5% for other passes. However, if the receiver received the ball in a positional way, at the height of his most direct opponent, the chances of success were reduced by 12%. Similar results to the variables studied can found for the space gain by confirmed attackers, moving away from the most direct defender, as has been shown in other studies [9,19]. Therefore, the player should opt for one of these two options to have greater probabilities of success, creating a useful space where he can obtain a greater advantage over rivals. 

On the other hand, a diagonal run by the receiver increased the chances of success by 7%. These results indicate greater effectiveness in counter-attacks or short possessions, as Sarmento et al. [45] concluded by analyzing a sample of 68 matches and 1694 offensive sequences; counter-attacks and rapid attacks increased the success of an offensive sequence by 40% compared to positional attacks. Lago Ballesteros et al. [29] analyzed 908 possessions obtained by a team in the Spanish soccer league in 12 matches and concluded that direct attacks and counter-attacks were three times more effective than positional attacks. Furthermore, Hughes and Franks [22] showed the proportion of goals to shots taken is better in “direct play” than in “possession play”. Consequently, with the data obtained in this the study and from the scientific literature, the way that a shorter duration of the play can be produced can be seen; that is, with faster attacks seeking to make a diagonal pass forward being received in a diagonal run.

The analysis of the reception space next to the path of the receiver shows that it seems to be more efficient to receive diagonally in separation. These results can be explained by the fact that a player receives at a point furthest away from his direct defender [44], looking for a free or useful space, and closer to the goal, thus obtaining a greater advantage to score a goal.

Regarding the receiver of the ball received in positional play, the play finished effectively 70.59% of the time and in a static position it finished effectively 58.36% of the time, which can explain why successful teams tend to play possession soccer to keep together lines in which there is not a large distance between their players. As already demonstrated by Kempe et al. [4] when they analyzed 676 Bundesliga games, successful teams prefer possession play and control of the game is the most important variable of success. With possession play, where passing sequences are longer and produce more goals per possession than shorter passing sequences [21], teams aim to seek out the free man and the space between the lines to create spaces that are occupied and generate advantages. In this way, a better team organization that generates a possibility of recovering the ball more quickly in the event of a loss is sought [28]. The combination of long possessions and short possessions seems to be a more effective way for a team to have control of the game and provides greater probabilities of scoring a goal, it being more effective to make a diagonal forward pass received in a diagonal run in separation.

On the other hand, the main indicator that delimits effectiveness in the completion of a play is making a successful pass [2]. When a very good pass is made there is difference of almost 14% in finishing the play effectively compared to passes made in a noneffective or neutral way. These data are in line with previous results of other studies in which successful passing improves effectiveness [2,19,34,40,49]. 

With regard to the zone of the field from where the pass was made, it should be noted that in zones 11 to 20 (pre-offensive zone in the right zone and offensive zone in the left zone), the probability of finishing the play successfully increased in comparison with when the finish was ineffective. However, zone 15 of the pitch was the only one in the pre-offensive zone where the success rate was reduced, by 0.6%. It was reduced due to the greater effectiveness of the equipment analysis in the central zone (zone 13) and the right zone (zone 11). On the other hand, zone 8 of the pitch, a pre-defensive zone in the central zone, had the greatest number of passes, followed by zone 13 (pre-defensive zone in the central part) and 11 (pre-defensive zone on the right). These results indicate the importance of reaching these zones when having possession of the ball, or stealing the ball in a forward zone and starting the play near the opponent’s goal. The results by zones show how passes in the pre-offensive and offensive zones have a 7% and 10% chance, respectively, of finishing the play successfully and this is reduced in the defensive and pre-defensive zones by 7% and 8%, respectively. Furthermore, it can be seen how more passes are made in the pre-offensive zone than in the rest of the zones. Also, reaching the offensive zone carries with it almost a 50% chance of finishing the play effectively. Tenga et al. [26] showed how more than half of goal opportunities and goals scored [27] started in the pre-offensive zone. Other authors showed that a higher proportion of possessions started in the pre-defensive and pre-offensive zones [10,11,50].

The effect of playing as a local or visitor did not show significant differences, which is in line with previous studies when measured in high-level teams [30,51]. However, the results show that when the match was played on the rival team’s pitch, there were almost 2% fewer passes but that there was a difference of 13% when the play ended effectively.

The state of the match influenced the percentage of possession because teams had more possession when they were losing matches than when they were winning or drawing, as was shown by an increase in possession by 1% every 11 min when they were losing [10,23], leading to a greater number of passes. 

In terms of the limitations of this study, the analysis of the reception space and the receiver’s trajectory could be more effectively deployed in combination with detailed analysis of individual parameter-like game events, such as passing networks, overtaking players, the centroid, dispersion and team synchronization. On the other hand, the analysis of two elite teams reflects their particular playing style, so care should be taken when extrapolating these results to other teams and contexts. 

## 5. Conclusions

The present study provided two novel variables for analyzing the effectiveness of play completion in elite soccer by analyzing the effects that the movements of players receiving passes had on a team’s attack. 

The space in which the pass was received, approach or separation, and the path of the receiver, diagonal or vertical, was positively related to the successful performance of the game. Also, the area of the pitch, pre-offensive or offensive, from which the passer made the pass influenced the effectiveness of the completion of the play. On the other hand, the diagonality of the pass was the variable studied that had the least relation to the efficiency of the play and thus could have greater relevance in the creation and progression of the play, generating useful spaces to be used by players.

The evidence provided in this study could help coaches and coaching staff to attend to the mobility of their players in attack, creating useful spaces to that allow the opposite goal to be reached with the highest probability of success. Also, exerting pressure on the opposing team in order to steal the ball in advanced areas or bring the ball as quickly as possible to areas close to the opponent’s area could be a decisive factor in the outcome of matches.

### Future Lines of Research

The results provided in this study can be analyzed together with other contextual variables, such as the speed of the receiver and the dispersion and synchronization of the equipment using the data provided by artificial intelligence. In addition, future research could be carried out on a sample in which different levels of equipment and different leagues are included to check potential differences.

## Figures and Tables

**Figure 1 ijerph-17-09396-f001:**
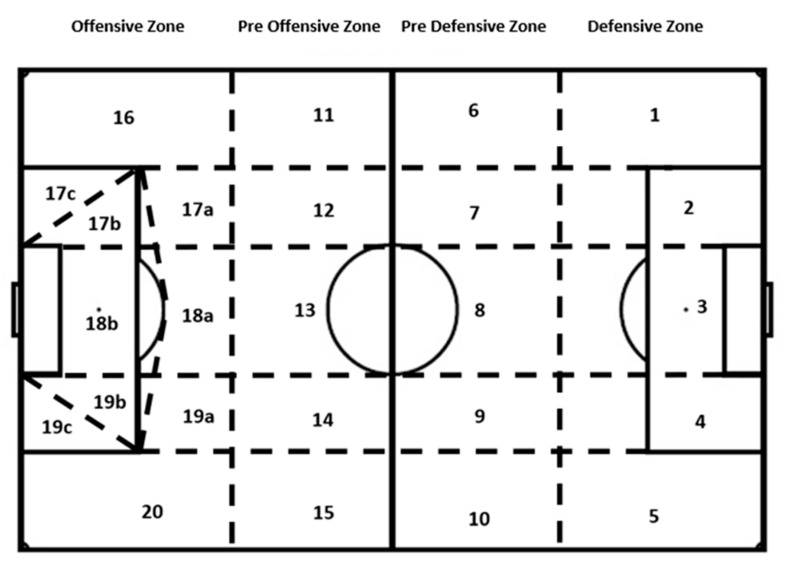
Sectorization of the soccer field into 20 zones. Modified from Fernandez-Navarro et al. (2016).

**Figure 2 ijerph-17-09396-f002:**
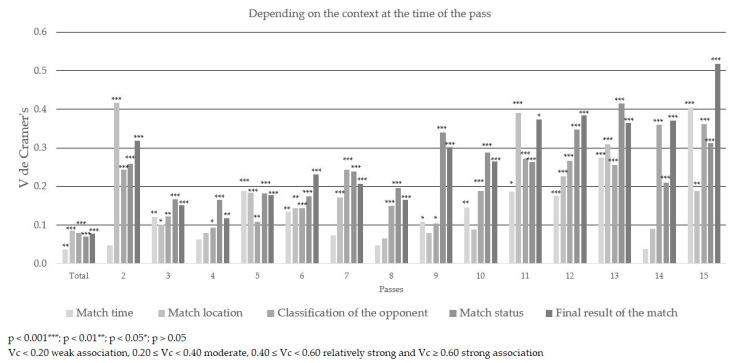
Association of the effectiveness of a pass (Vc) and the level of significance (*p*) according to the completion of the play.

**Figure 3 ijerph-17-09396-f003:**
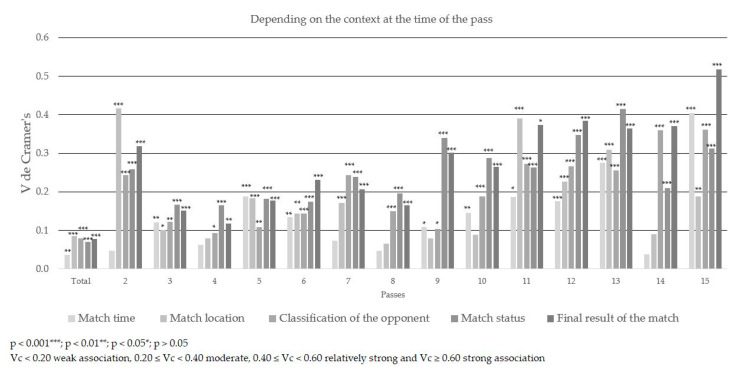
Association of the categories of the context at the time of the pass (Vc) and the level of significance (*p*) according to the completion of the play.

**Figure 4 ijerph-17-09396-f004:**
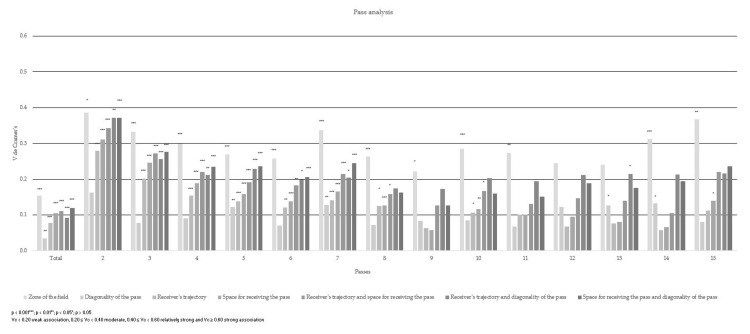
Association of the categories of pass analysis (Vc) and the level of significance (*p*) according to the completion of the play.

**Table 1 ijerph-17-09396-t001:** Categories and dimensions of the action of the pass analyzed.

Categories and Dimensions Depending on the Context at the Time of the Pass
Match time
1st part: playing time from the referee’s whistle at the beginning of the first part until the referee’s whistle at the end of the first part.
2nd part: playing time from the referee’s whistle at the beginning of the second part until the end of the match.
Match location
Home: the match is played on the pitch of the analyzed team.
Away: the match is played in the pitch of the opposing team.
Classification of the opponent
Group 1: from 1st to 5th place in La Liga standings.
Group 2: from 6th to 10th place in La Liga standings.
Group 3: 11th to 15th place in La Liga standings.
Group 4: 16th to 20th place in La Liga standings.
Final result of the match
Win by >1 goal: the team observed scored two or more goals than the opponent.
Win by 1 goal: the team observed scored one goal more than the opponent.
Tie: the observed team scored the same number of goals as the opponent.
Loss by 1 goal: the observed team scored one goal less than the opponent.
Loss by >1 goal: the observed team scored two or more goals less than the opponent.
Match status
Winning by >1 goal: the observed team scored two or more goals than the opponent.
Winning by 1 goal: the observed team scored one more goal than the opponent.
Tying: the observed team scored the same number of goals as the opponent.
Losing by 1 goal: the observed team scored one goal less than the opponent.
Losing by >1 goal: the observed team scored two or less goals than the opponent.
Categories and dimensions in performance analysis
Effectiveness of completion
Effective: shot that ends in goal, shot on goal, shot defended by the goalkeeper, shot out, shot against the opponent, direct free kick, corner, penalty and shot to the center of the area.
Neutral: maintenance of possession by the observed team (from a throw-in, long foul or other situations).
Ineffective: ball recovery by the opponent, ball out, end of possession for violation of the rules of the game.
Effectiveness of pass
Very good: overpassed rival players and gave benefit to the receiver.
Good pass: received by the teammate and possession was maintained.
Bad pass: loss of the ball.
Categories and dimensions in pass analysis
Zone of the field
Defensive zone: zone 1, 2, 3, 4 and 5.
Pre-defensive zone: zone 6, 7, 8, 9 and 10.
Pre offensive zone: zone 11, 12, 13, 14 and 15.
Offensive zone: zone 16, 17a, 17b, 17c, 18a, 18b, 19a, 19b, 19c and 20.
Diagonality of the pass
In front: every time the player in possession made a pass towards the opposite goal.
At the back: every time the player in possession made a pass towards the defended goal.
At the side: each time the player in possession made a side pass to the axis of attack.
Diagonal forward: every time the player in possession made a pass diagonally to the axis of attack towards the opposite goal (the ball moved towards the corridor and sector).
Diagonal backward: every time the player in possession made a pass diagonally to the axis of attack towards the defended goal (the ball moved towards the corridor and sector).
Receiver’s trajectory
Diagonal: the receiver of the pass received the ball running diagonally to his starting position in the race and the pitch.
Vertical: the receiver of the pass received the ball running vertically with respect to his starting position in the race and the field of play.
Perpendicular: the receiver of the pass received the ball running perpendicular to his starting position in the race and the field of play.
Static: the receiver of the pass received the ball without being on the run.
Space for receiving the pass
Approach: the player acquired positional superiority by placing himself between intervals on a trajectory approaching the ball’s passer.
Separation: the player acquired positional superiority by placing himself between intervals on a trajectory away from the passer of the ball.
Positional: the player received the ball at the height of his most direct opponent.

**Table 2 ijerph-17-09396-t002:** Intra- and inter-observer Kappa index values.

Categories	n	K Inter-Observer	n	K Intra-Observer
Time of the pass				
Match time	920	1.000	1371	1.000
Match location	920	1.000	1371	1.000
Classification of the opponent	920	1.000	1371	1.000
Final result of the match	920	1.000	1371	1.000
Match status	920	1.000	1371	1.000
Performance analysis				
Effectiveness of completion	920	1.000	1371	1.000
Effectiveness of pass	920	0.980	1371	1.000
Pass analysis				
Zone of the field	920	0.992	1371	1.000
Diagonality of the pass	920	0.994	1371	0.893
Receiver’s trajectory	920	0.978	1371	0.917
Space for receiving the pass	920	0.976	1371	0.886

**Table 3 ijerph-17-09396-t003:** Descriptive statistics of the total number of passes for the categories and dimensions showing sample (n) and percentage (%).

Categories and Dimension		Total							
		Ineffective		Neutral		Effective		Total	
		n	%	n	%	n	%	n	%
Time									
	1st part	3199	53.67	607	55.89	1554	50.44	5360	52.92
	2nd part	2762	46.33	479	44.11	1527	49.56	4768	47.08
Location									
	Home	2947	49.44	466	42.91	1745	56.64	5158	50.93
	Away	3014	50.56	620	57.09	1336	43.36	4970	49.07
Classification of the opponent									
	Group 1	1117	18.74	106	9.76	614	19.93	1837	18.14
	Group 2	1440	24.16	236	21.73	729	23.66	2405	23.75
	Group 3	2162	36.27	461	42.45	1274	41.35	3897	38.48
	Group 4	1242	20.84	283	26.06	464	15.06	1989	19.64
Final result of the match									
	Lose > 1 goal	762	12.78	168	15.47	306	9.93	1236	12.20
	Lose by 1 goal	402	6.74	118	10.87	344	11.17	864	8.53
	Tie	1700	28.52	262	24.13	685	22.23	2647	26.14
	Win by 1 goal	1335	22.40	238	21.92	772	25.06	2345	23.15
	Win > 1 goal	1762	29.56	300	27.62	974	31.61	3036	29.98
Match status									
	Losing > 1 goal	109	1.83	62	5.71	102	3.31	273	2.70
	Losing by 1 goal	629	10.55	139	12.80	400	12.98	1168	11.53
	Tying	4006	67.20	653	60.13	1933	62.74	6592	65.09
	Winning by 1 goal	895	15.01	143	13.17	425	13.79	1463	14.45
	Winning > 1 goal	322	5.40	89	8.20	221	7.17	632	6.24
Effectiveness of pass									
	Bad	649	10.89	43	3.96	3	0.10	695	6.86
	Good	5064	84.95	1003	92.36	2533	82.21	8600	84.91
	Very good	248	4.16	40	3.68	545	17.69	833	8.22
Diagonality of the pass									
	In front	498	8.35	95	8.75	211	6.85	804	7.94
	At the back	372	6.24	59	5.43	167	5.42	598	5.90
	At the side	945	15.85	194	17.86	497	16.13	1636	16.15
	Diagonal forward	2456	41.20	432	39.78	1398	45.37	4286	42.32
	Diagonal backward	1690	28.35	306	28.18	808	26.23	2804	27.69
Receiver’s trajectory									
	Diagonal	905	15.18	134	12.34	684	22.20	1723	17.01
	Vertical	787	13.20	154	14.18	483	15.68	1424	14.06
	Perpendicular	186	3.12	30	2.76	116	3.77	332	3.28
	Static	4083	68.50	768	70.72	1798	58.36	6649	65.65
Space for receiving the pass									
	Approach	474	7.95	65	5.99	402	13.05	941	9.29
	Separation	553	9.28	79	7.27	504	16.36	1136	11.22
	Positional	4934	82.77	942	86.74	2175	70.59	8051	79.49
Total		5961	100	1086	100	3081	100	10128	100

**Table 4 ijerph-17-09396-t004:** Descriptive statistics showing sample (n) and percentage (%) of passes for each zone of the field.

Zone of the Field	Ineffective		Neutral		Effective		Total	
	n	%	n	%	n	%	n	%
1	165	2.77	52	4.79	48	1.56	265	2.62
2	146	2.45	30	2.76	30	0.97	206	2.03
3	375	6.29	83	7.64	136	4.41	594	5.86
4	192	3.22	40	3.68	53	1.72	285	2.81
5	176	2.95	41	3.78	56	1.82	273	2.70
6	360	6.04	73	6.72	122	3.96	555	5.48
7	393	6.59	69	6.35	158	5.13	620	6.12
8	619	10.38	107	9.85	222	7.21	948	9.36
9	377	6.32	84	7.73	166	5.39	627	6.19
10	383	6.43	59	5.43	140	4.54	582	5.75
11	436	7.31	79	7.27	312	10.13	827	8.17
12	392	6.58	56	5.16	237	7.69	685	6.76
13	468	7.85	60	5.52	302	9.80	830	8.20
14	395	6.63	83	7.64	258	8.37	736	7.27
15	473	7.93	93	8.56	224	7.27	790	7.80
16	167	2.80	17	1.57	142	4.61	326	3.22
17a	67	1.12	5	0.46	70	2.27	142	1.40
17b	11	0.18	1	0.09	33	1.07	45	0.44
17c	5	0.08	1	0.09	17	0.55	23	0.23
18a	57	0.96	4	0.37	50	1.62	111	1.10
18b	7	0.12	0	0.00	28	0.91	35	0.35
19a	77	1.29	11	1.01	77	2.50	165	1.63
19b	21	0.35	6	0.55	29	0.94	56	0.55
19c	3	0.05	0	0.00	8	0.26	11	0.11
20	196	3.29	32	2.95	163	5.29	391	3.86
Total	5961	100	1086	100	3081	100	10128	100

**Table 5 ijerph-17-09396-t005:** Descriptive statistics by area of the course showing sample (n) and percentage (%) of passes.

Zone of the Field		Ineffective		Neutral		Effective		Total	
		n	%	n	%	n	%	n	%
	Defensive	1054	17.68	246	22.65	323	10.48	1623	16.02
	Pre-defensive	2132	35.77	392	36.10	808	26.23	3332	32.90
	Pre-offensive	2164	36.30	371	34.16	1333	43.27	3868	38.19
	Offensive	611	10.25	77	7.09	617	20.03	1305	12.89
Total		5961	100	1086	100	3081	100	10128	100

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
