# Peer review of "What Is the Relevance in the Passing Action between the Passer and the Receiver in Soccer? Study of Elite Soccer in La Liga"

_ijerph, 2020, doi:10.3390/ijerph17249396_

Round 1

Reviewer 1 Report

Title.

What is the Relevance in the Passing Action between 2 the Passer and the Receiver in Soccer? Study of Elite Soccer in La Liga

To carry out the study, the pass variable was analyzed in 20 soccer matches of the 2018/2019 season of La Liga (Spanish First soccer league) using video recordings.

According to the authors, the main aim of this study was to analyses the pass regarding to the trajectory made by the receiver of the ball and the space where player received it in relation to the closest direct defender and his efficiency in finishing the play.

General comments

This study involves a very hard data analysis and must be recognized as such. The analysis procedure is adequate, and the reliability of the evaluator is assessed. The limitations on the characteristics of the soccer teams analysed are appropriate. However, the weak part of the study, in our opinion, is that the conclusions do not allow for much improvement in the knowledge of soccer tactics. In this sense, the practical applications seem to be evident before the study is carried out. However, we are aware of how extremely difficult it is to improve knowledge in this area.

I do not have important considerations about the manuscript, but I do have some minor revisions that I present below.

Minor revisions

Line 268. “The main data showed the relationship between the trajectory of the receiver and the space where he receives the ball, and demonstrates that…”

The term "demonstrates" should be replaced by "suggests that" or "shows that" or similar term, because in the social sciences, and especially in soccer, nothing is "demonstrated", let alone based on correlations.

Lines 270-71 “However, if the receiver receives the ball in a positional way…”

The term "positional way" should be clarified in this context

Lines 339-40 “The space in which the pass is received and the career path of the receiver is positively related to the successful performance of the game… Also, the area of the pitch”

The specific space should be indicated, and also the area of the pitch

Author Response

General considerations

Firstly, we would like to thank reviewers and associate editor their comments, which, undoubtedly, have contributed to correct some limitations in the manuscript and therefore improve its quality. Important changes have been done following their suggestions. Significant changes made in the manuscript have been marked in red color.

We shall now proceed to individually answer each reviewer comments. Black color will be used to introduce reviewers’ comments and blue will be used in our responses. Changes in the manuscript have been highlighted in red color.

Reviewer 1

General comments

This study involves a very hard data analysis and must be recognized as such. The analysis procedure is adequate, and the reliability of the evaluator is assessed. The limitations on the characteristics of the soccer teams analysed are appropriate. However, the weak part of the study, in our opinion, is that the conclusions do not allow for much improvement in the knowledge of soccer tactics. In this sense, the practical applications seem to be evident before the study is carried out. However, we are aware of how extremely difficult it is to improve knowledge in this area.

I do not have important considerations about the manuscript, but I do have some minor revisions that I present below.

Minor revisions

Line 268. “The main data showed the relationship between the trajectory of the receiver and the space where he receives the ball, and demonstrates that…”

The term "demonstrates" should be replaced by "suggests that" or "shows that" or similar term, because in the social sciences, and especially in soccer, nothing is "demonstrated", let alone based on correlations.

We have changed the term demonstrates to suggest because it is true that we rely on correlations in football. Thank you for the suggestion in the terminology used.

The main data showed the relationship between the trajectory of the receiver and the space where he receives the ball, and suggests that if the receiver takes…

Lines 270-71 “However, if the receiver receives the ball in a positional way…”

The term "positional way" should be clarified in this context

We have specified that it means positional way because it is a specific term used for where the ball is made the receiver. The definition of positional way is the player receives the ball at the height of his most direct opponent.

However, if the receiver receives the ball in a positional way, at the height of his most direct opponent…

Lines 339-40 “The space in which the pass is received and the career path of the receiver is positively related to the successful performance of the game… Also, the area of the pitch”

The specific space should be indicated, and also the area of the pitch

We have solved this problem just like the previous one. We have specified which space and part of the field it is.

The space in which the pass is received, approach or separation, and the career path of the receiver, diagonal or vertical, is positively related to the successful performance of the game. Also, the area of the pitch, pre-offensive or offensive…

Thank you for the last two corrections that have made us specify the terminology used for better understanding by the reader.

Reviewer 2 Report

The authors have several mistakes during the manuscript. A version with corrections are attached to the report. The authors may rewrite the introduction. There's information about assessing methods that are not used on the methodology and not presented in results. The results must be reorganized with descriptives first.

This paper require a lot of changes.

Reviewer 3 Report

I thank the authors for presenting the results of their research. I realize how much work had to be done to analyze so many matches and evaluate so many parameters.
I believe that a very interesting piece of work has been created, allowing for a detailed analysis of the playing style of these two top teams, not only in the Spanish league (La Liga), but also in the European League.

One small note:
As Table 4 contains a lot of interesting results, and at the same time is not readable, I suggest to the Authors that instead of the table, introduce a panel with bar graphs, which will present the values of Cramer's coefficients with horizontal lines showing the size of the effect. It is also worth adding symbols that indicate the level of significance. I prepared a sample of one of these graphs, it is attached. In my opinion, this solution will be much clearer and will enrich this work.

Round 2

Reviewer 2 Report

The authors have made a substantial effort to improve the manuscript.

In my opinion it is suitable for publication.